# An Integrated Single-Beam Three-Axis High-Sensitivity Magnetometer

**DOI:** 10.3390/s23063148

**Published:** 2023-03-15

**Authors:** Shengran Su, Zhenyuan Xu, Xiang He, Chanling Yin, Miao Kong, Xuyuan Zhang, Yi Ruan, Kan Li, Qiang Lin

**Affiliations:** 1Zhejiang Provincial Key Laboratory for Quantum Precision Measurement, College of Science, Zhejiang University of Technology, Hangzhou 310023, China; 2Hangzhou Qmag Technology Co., Ltd., Hangzhou 310023, China

**Keywords:** three-axis magnetometer, bio-magnetic measurement, high-sensitivity magnetometer

## Abstract

Three-axis atomic magnetometers have great advantages for interpreting information conveyed by magnetic fields. Here, we demonstrate a compact construction of a three-axis vector atomic magnetometer. The magnetometer is operated with a single laser beam and with a specially designed triangular ${}^{87}\mathrm{Rb}$ vapor cell (side length is 5 mm). The ability of three-axis measurement is realized by reflecting the light beam in the cell chamber under high pressure, so that the atoms before and after reflection are polarized along two different directions. It achieves a sensitivity of 40 $fT/\sqrt{Hz}$ in x-axis, 20 $fT/\sqrt{Hz}$ in y-axis, and 30 $fT/\sqrt{Hz}$ in z-axis under spin-exchange relaxation-free regime. The crosstalk effect between different axes is proven to be little in this configuration. The sensor configuration here is expected to form further values, especially for vector biomagnetism measurement, clinical diagnosis, and field source reconstruction.

## 1. Introduction

Atomic magnetometers (AM) have seen rapid development and have shown great potential in widespread applications ranging from fundamental physics [1] and biomedicine [2,3] to geophysics [4] and industrial inspection [5,6]. Without the requirement of cryogenics, AMs have become viable high-sensitivity alternatives to the magnetic sensor based on the superconducting quantum interference device (SQUID), which is currently commercially available.

Various configurations of AMs have been proposed and realized to meet specific requirements in practical applications. Among them, AMs operated in the condition of the spin-exchange relaxation-free (SERF) regime enable the highest sensitivity [7], comparable to SQUID magnetometers. This has attracted great attention in bio-magnetic measurements, such as magnetoencephalography (MEG) and magnetocardiography (MCG). As well as reducing manufacturing and operating costs compared to SQUID magnetometer, using AMs to do bio-magnetic measurements has many advantages. Taking the MEG as an example, the shape of the human skull varies with each individual, but in SQUID MEG equipment, the overall shape of the sensor probe and the distribution of the discrete pick-up coils are fixed due to the inescapable Dewar flask. This precludes fully exerting the high sensitivity of the SQUID magnetometer. However, for room-temperature operable AM sensors, a high degree of miniaturization and integration allows for a flexible design of the sensor array in the form of a helmet custom-made according to individual skull shape.

Vector property is one of the main driving forces for extracting information about the object from its magnetic field. Especially for biomagnetism, vector measurement provides more information for the source inversion problem or focal diagnosis. The accuracy of spatial inversion can be improved using the direction information of the magnetic field [8]. Great efforts have been made to realize three-axis vector AMs. For example, we can gain vector information using multiple circularly polarized laser beams [9], or using the AC Stark shift [10], electromagnetically induced transparency resonances [11] and atomic alignment phenomenon [12]. Typically, SERF AM only responds to the biaxial magnetic fields that are perpendicular to the direction of light propagation. By modulating magnetic fields in the two insensitive directions, Seltzer et al. introduced a triaxial SERF magnetometer working in an unshielded environment [13]. Two laser beams of different wavelengths were required in this configuration, which was unfavorable for the miniaturization of sensors. Further introducing modulation fields to all three directions, a scheme for constructing a compact vector SERF AM with a single beam was proposed in [14], where the sensitivity of one axis (3000 $fT/\sqrt{Hz}$) was significantly worse than the other two axes (350 $fT/\sqrt{Hz}$). Junjian Tang et al. realized a single-beam vectorial AM by applying a small DC offset field and a high-frequency modulation field [15]. In this way, the crosstalk effect between magnetic fields in different directions is enhanced for the magnetometer and therefore enables the magnetometer to respond to changes in magnetic fields in three directions. Recently, a vectorial AM was improved from the typical biaxial SERF and used for multi-channel MEG [16]. In this scheme, one laser beam was divided into two parts. The post-split beams passed through the vapor cell from different directions and interacted with different parts of atoms. Then, two photodetectors collected the beams. Two beams in such a small cell (3 mm × 3 mm × 3 mm, inside) should avoid overlapping in space, and the effective atom number for sensing magnetic fields on each axis is reduced.

Here, a new compact triaxial SERF AM is proposed and experimentally demonstrated. A triangular prism-shaped vapor cell is used as the sensing element. Vectorial measurement is achieved by reflecting a single beam once inside the cell and, respectively, applying modulation magnetic fields to all three axes. Only one photodetector is required in the scheme. The atomic spin dynamics are numerically investigated to understand the principle of the AM. In the experiment, vectorial measurement of magnetic fields is performed with high sensitivity in all three axes (40 $fT/\sqrt{Hz}$ in x-axis, 20 $fT/\sqrt{Hz}$ in y-axis, and 30 $fT/\sqrt{Hz}$ in z-axis). The crosstalk effect between different axes is proven to be small. The AM configuration shown here is suitable for bio-magnetic measurement.

## 2. Theory and Numerical Analysis

### 2.1. Setup

The schematic diagram is shown in Figure 1. We use a triangular prism-shaped vapor cell with a side length of 0.5 cm, which contains a drop of enriched ${}^{87}Rb$ atoms and 760 Torr of nitrogen as quenching and buffer gas. A miniature low-noise heater is attached to the cell wall, which heats the cell to 150 °C using a high-frequency alternating current. A circularly polarized light beam with a wavelength of 795 nm and a diameter of around 2 mm, emitted from a single-mode vertical-cavity surface-emitting laser (VCSEL), is incident on one side of the cell and reflected by one polarization-maintaining mirror, which is attached to the bevel of the cell. Three pairs of coils are mounted around the cell to compensate for the background magnetic field and applying modulation fields along three orthogonal axes. When the AM starts to work, an automated program is run to realize the compensation process and optimization of the wavelength of the light. Close to the exit surface of the light, one single photodetector is used to monitor light absorption. Benefiting from the simple optics system without any beam-splitting element and the adoption of VCSEL as the light source, the whole sensor is highly integrated (Figure 1). In our experiments, the sensor is placed inside a five-layer magnetic shield.

### 2.2. Principle of the AM

By reflecting the light beam, the atoms are polarized along two orthogonal directions. A mass of buffer gas greatly shortens the mean free path of ${}^{87}Rb$ atoms, making the reflection surface a boundary of atoms with different magnetization vectors. Interacting with the circularly polarized light, ${}^{87}Rb$ atoms in the cell can be divided into two parts—P1 and P2—which are polarized in two mutually perpendicular directions, x and z. In the experiment, a modulation field ${\hat{B}}^{m}=B_{x}^{m}\cos\left(2\omega t\right)\hat{x}+B_{y}^{m}\cos\left(\omega t\right)\hat{y}+B_{z}^{m}\sin\left(\omega t\right)\hat{z}$ is applied. It is worth noting that, in order to avoid frequency noise during signal extraction, the modulation frequency in the x direction is selected as twice the modulation frequency in the y and z directions. For the first part of the atoms, the Bloch equation, describing the dynamics of the atomic polarization *P*, can be expressed as
(1)$$\frac{d}{dt}P^{1}=\;\frac{1}{q}\left[\gamma B\times P^{1}+R_{op}\left(s\hat{x}-P^{1}\right)-R_{rel}P^{1}\right],$$
where *q* is the nuclear slowing factor, γ is the gyromagnetic ratio, $R_{op}$ is the pumping rate, *s* is the photon polarization of the pump beam, and $R_{rel}$ is the total spin-relaxation rate.

By setting $\frac{d}{dt}P=0$, we can obtain the steady-state solution to Equation (Equation 1) as
(2)$$\begin{array}{c} P_{x}^{1}=P_{0}^{1}\frac{1+{\beta_{1}}^{2}{B_{x}}^{2}}{1+{\beta_{1}}^{2}{B_{x}}^{2}+\beta_{1}^{2}{B_{y}}^{2}+{\beta_{1}}^{2}{B_{z}}^{2}},\\ P_{y}^{1}=P_{0}^{1}\frac{-\beta_{1}B_{z}+{\beta_{1}}^{2}B_{x}B_{y}}{1+{\beta_{1}}^{2}{B_{x}}^{2}+\beta_{1}^{2}{B_{y}}^{2}+{\beta_{1}}^{2}{B_{z}}^{2}},\\ P_{z}^{1}=P_{0}^{1}\frac{\beta_{1}B_{y}+{\beta_{1}}^{2}B_{x}B_{y}}{1+{\beta_{1}}^{2}{B_{x}}^{2}+\beta_{1}^{2}{B_{y}}^{2}+{\beta_{1}}^{2}{B_{z}}^{2}}, \end{array}$$
where $B_{x}=B_{x}^{0}+B_{x}^{m}\cos\left(2\omega t\right)$, $B_{y}=B_{y}^{0}+B_{y}^{m}\cos\left(\omega t\right)$ and $B_{z}=B_{z}^{0}+B_{z}^{m}\sin\left(\omega t\right)$. $B_{x(y,z)}^{0}$ is the component of the static magnetic field projected along the corresponding direction. $\beta =\frac{\gamma }{R_{op}+R_{rel}}$, and $P_{0}=\frac{sR_{op}}{R_{op}+R_{rel}}$ represents the degree of atomic polarization in the absence of any magnetic field. Superscripts are used to distinguish atoms of P1 part and P2 part.

Similarly, for atoms of the P2 part, the atomic polarization along the direction of light propagation is
(3)$$P_{z}^{2}=P_{0}^{2}\frac{1+{\beta_{2}}^{2}{B_{z}}^{2}}{1+{\beta_{2}}^{2}{B_{x}}^{2}+{\beta_{2}}^{2}{B_{y}}^{2}+{\beta_{2}}^{2}{B_{z}}^{2}}.$$
Atomic polarization is measured by optical absorption, which is inversely correlative with the atomic polarization along the light-propagation direction. Therefore, the joint contribution of the two parts of atoms can be written as
(4)$$P_{probe}=P_{x}^{1}+P_{z}^{2}.$$

Ignoring the higher-order term and keeping the zero-order and the first-order term of the Taylor expansion of $P_{probe}$, we obtain
(5)$$\begin{array}{cc} \; & P_{probe}=P_{0}^{1}+P_{0}^{2}+\\ \; & \left\{-\frac{{{\beta_{1}}^{4}P}_{0}^{1}\left({B_{y}}^{2}+{B_{z}}^{2}\right)}{{\left(1+{\beta_{1}}^{2}{B_{x}}^{2}+\beta_{1}^{2}{B_{y}}^{2}+{\beta_{1}}^{2}{B_{z}}^{2}\right)}^{2}}+\frac{{\beta_{2}}^{2}P_{0}^{2}\left(1+{\beta_{2}}^{2}{B_{z}}^{2}\right)}{{\left(1+{\beta_{2}}^{2}{B_{x}}^{2}+\beta_{2}^{2}{B_{y}}^{2}+{\beta_{2}}^{2}{B_{z}}^{2}\right)}^{2}}\right\}{2B}_{x}B_{x}^{m}\sin\left(2\omega t\right)\\ \; & +\left\{\frac{{\beta_{1}}^{2}P_{0}^{1}\left(1+{\beta_{1}}^{2}{B_{x}}^{2}\right)}{{\left(1+{\beta_{1}}^{2}{B_{x}}^{2}+\beta_{1}^{2}{B_{y}}^{2}+{\beta_{1}}^{2}{B_{z}}^{2}\right)}^{2}}+\frac{{\beta_{2}}^{2}P_{0}^{2}\left(1+{\beta_{2}}^{2}{B_{z}}^{2}\right)}{{\left(1+{\beta_{2}}^{2}{B_{x}}^{2}+\beta_{2}^{2}{B_{y}}^{2}+{\beta_{2}}^{2}{B_{z}}^{2}\right)}^{2}}\right\}{2B}_{y}B_{y}^{m}\sin\left(\omega t\right)\\ \; & +\left\{-\frac{{\beta_{1}}^{2}P_{0}^{1}\left(1+{\beta_{1}}^{2}{B_{x}}^{2}\right)}{{\left(1+{\beta_{1}}^{2}{B_{x}}^{2}+\beta_{1}^{2}{B_{y}}^{2}+{\beta_{1}}^{2}{B_{z}}^{2}\right)}^{2}}+\frac{{{\beta_{2}}^{4}P}_{0}^{2}\left({B_{x}}^{2}+{B_{y}}^{2}\right)}{{\left(1+{\beta_{2}}^{2}{B_{x}}^{2}+\beta_{2}^{2}{B_{y}}^{2}+{\beta_{2}}^{2}{B_{z}}^{2}\right)}^{2}}\right\}{2B}_{z}B_{z}^{m}\cos\left(\omega t\right). \end{array}$$
When circularly polarized light passes through the P1 part, the light is absorbed by atoms and its intensity is weakened, making $P_{0}^{1}$ and $P_{0}^{2}$ different ($P_{0}^{1}>P_{0}^{2}$). This also makes a slight difference between $\beta_{1}$ and $\beta_{2}$. When the background magnetic field is well compensated to zero, and the modulation field is weak, ignoring the difference between $\beta_{1}$, $\beta_{2}$, Equation (Equation 5) can be simplified as
(6)$$\begin{array}{c} {P_{probe}}^{1}\propto P_{0}^{2}B_{x}^{m}B_{x}^{0}\sin\left(2\omega t\right)+\left(P_{0}^{1}+P_{0}^{2}\right)B_{y}^{m}B_{y}^{0}\sin\left(\omega t\right)-P_{0}^{1}{B_{z}^{0}B}_{z}^{m}\cos\left(\omega t\right). \end{array}$$
We can see that the values of the three-axis magnetostatic field $B_{x}^{0}$, $B_{y}^{0}$ and $B_{z}^{0}$ can be obtained by, respectively, extracting the signal components of $\sin\left(2\omega t\right)$, $\sin\left(\omega t\right)$, and $\cos\left(\omega t\right)$ through the lock-in amplifier.

### 2.3. Numerical Analysis

Two approximations are introduced for the analytical solution (Equation (Equation 6)). First, $\frac{d}{dt}P=0$ must be held, meaning that the modulation frequency should be very low. Second, the static magnetic field $B^{0}$ and modulation field $B^{m}$ should be weak. However, low modulation frequency leads to narrow measurement bandwidth of AM, and low modulation amplitude leads to weak signal and thus low sensitivity. To understand the mechanism of our AM under more general conditions, we do the numerical analysis of Equation (Equation 1). The comparison between the analytic and numerical results is shown in Figure 2. Figure 2a,b are the results under the low-frequency weak magnetic field modulation ($f_{x}=20$ Hz, $f_{y}=10$ Hz, $f_{z}=10$ Hz, $B_{x}^{m}=1$ nT, $B_{y}^{m}=0.5$ nT, $B_{z}^{m}=0.5$ nT) and high-frequency strong magnetic field modulation ($f_{x}=2$ kHz, $f_{y}=1$ kHz, $f_{z}=1$ kHz, $B_{x}^{m}=100$ nT, $B_{y}^{m}=50$ nT, $B_{z}^{m}=50$ nT), respectively. Both of them are calculated under the assumption of $B_{x}^{0}=1$ nT, $B_{y}^{0}=1$ nT, $B_{z}^{0}=1$ nT, $R_{op}=300$ Hz and $R_{op}+R_{rel}=1000$ Hz. The initial polarization is set $P0_{x}=0.3$, $P0_{y}=0$ and $P0_{z}=0$. The dynamics of the polarization within the period of ten complete cycles are presented for both two cases, i.e., $t_{0}=1$ s →$t_{1}=2$ s for the case of weak modulation and $t_{0}=10$ ms →$t_{1}=20$ ms for the case of strong modulation. We can see that the polarization trajectories of ten cycles are perfectly overlapped, meaning steady states have been reached during the periods shown in Figure 2. The analytic result agrees well with the numerical one under weak modulation, while obvious deviation between them is observed under strong modulation.

Under the condition of strong modulation, in Figure 3, we plot the responses of the atomic spin to a change of 1 nT in the static magnetic field in three directions with the numerical method. The change in the trajectory of the atomic polarization vector proves that the atoms of P1 part are insensitive to the magnetic field change in the x direction, while being sensitive to the changes in the y and z directions. Similarly, for atoms of P2 part, its polarization vector is insensitive to the change in the z direction, while sensitive to the changes in the x and y directions. Therefore, the combination of atoms P1 and atoms P2 constitutes a three-axis vector magnetometer. To investigate the crosstalk effect among the measurements in different directions, we also numerically simulate the AM response to a change of 200 pT in the static magnetic field with a digital lock-in amplifier program. From the result shown in Figure 3c, it is found that the crosstalk effect among measurements of different directions is very small, and can be ignored. The triaxial responses are not the same, leading to different sensitivities in different directions. According to Equation (Equation 6), the response in the y direction is the largest as both parts of atoms contribute to the signal. The response in the x direction is smaller than in the z direction because $P_{0}^{1}>P_{0}^{2}$.

## 3. Experimental Results

In the experiment, we successively applied a 200 pT magnetic field with a duration of 5 s along the x, y, and z axes, and measured the response signals of the AM with a triaxial-signal synchronous acquisition system. Here, the modulation frequencies are chosen as 2000 Hz, 1000 Hz and 1000 Hz for x, y and z directions, respectively. The amplitudes of the modulation fields are chosen as 100 nT, 50 nT and 50 nT for the x, y and z directions, respectively. The recorded response signals are shown in Figure 4a. As expected in the theoretical analysis of previous sections, the responses of the AM to the magnetic fields of the same magnitude in three directions are different. The crosstalk characteristics could also be observed in Figure 4a. The quantified normalized responses in the x, y and z directions by the crosstalk effect are listed in Table 1, in which the all the crosstalk responses were lower than 3%.

The response signal is the strongest to the magnetic field along the y direction and is the weakest to the magnetic field along the x direction. It thus leads to different sensitivities related to different directions. As shown in Figure 4b, We achieved triaxial sensitivity of 40 $fT/\sqrt{Hz}$ in the x direction, 20 $fT/\sqrt{Hz}$ in the y direction and 30 $fT/\sqrt{Hz}$ in the z direction. The response result (Figure 4a) also indicates that the crosstalk effect among measurements of different directions is small. The frequency response measurement result is shown in Figure 4c. The 3 dB bandwidths were 80 Hz, 90 Hz and 80 Hz for x, y and z directions.

We also verified the measurement accuracy of the vectorial AM. We applied magnetic fields with magnitudes of 400 pT with various spatial orientations. The magnetic fields are generated with three pairs of mutually orthogonal coils. The relationships between the magnetic field and the input current of the coils have been calibrated. The comparison between the experimental and expected values of the applied magnetic fields is shown on a sphere (Figure 5). The experimental results are marked with blue dots on the sphere. For each dot, 2000 times of measurements have been performed for an estimation of the magnetic field. The projections of each dot on each coordinate plane are also presented. We can see the accuracy of the AM is high. The average angle error of 20 measurement points is 1.2°, and the value error is 1.3%. As well as the measurement error contributed by the AM itself, the error may be introduced by the non-orthogonality of the coils and the inaccuracy of the coil calibration. The standard deviation of 2000 measurements for each dot is shown in Figure 6, indicating that the AM is stable and has good repeatability.

## 4. Conclusions

Here, a novel scheme of a vectorial atomic magnetometer is described and experimentally verified. Highly sensitive vectorial measurement is realized by reflecting the light beam inside the cell in the SERF regime. Our scheme relaxes the requirement for the light and photodetector, facilitating the realization of a compact vectorial magnetic sensor or a sensor array. It has great potential in biomagnetism measurements, such as MCG and MEG.

## Figures and Tables

**Figure 1 sensors-23-03148-f001:**
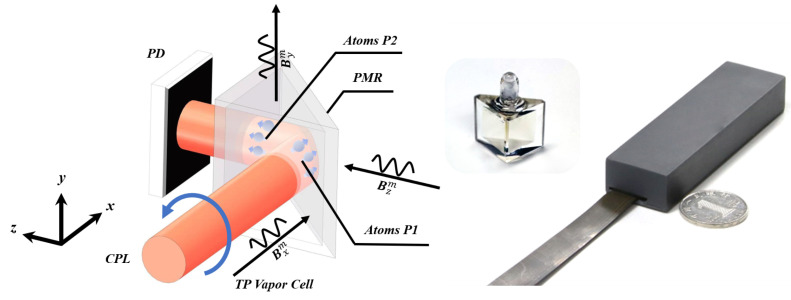
Schematic diagram for the triaxial magnetic field measurement. PMR: Polarization-maintaining reflector. CPL: Circularly polarized light. PD: Photodetector. The sensor and vapor cell images are on the right.

**Figure 2 sensors-23-03148-f002:**
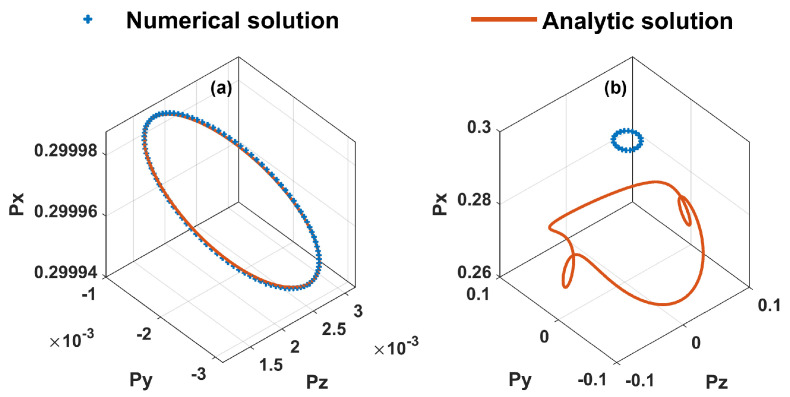
Comparison between the analytic and numerical results of the dynamic of the atomic polarization vector within the period of ten complete cycles under (**a**) weak modulation ($f_{x}=20$ Hz, $f_{y}=10$ Hz, $f_{z}=10$ Hz, $B_{x}^{m}=1$ nT, $B_{y}^{m}=0.5$ nT, $B_{z}^{m}=0.5$ nT) and (**b**) strong modulation ($f_{x}=2$ kHz, $f_{y}=1$ kHz, $f_{z}=1$ kHz, $B_{x}^{m}=100$ nT, $B_{y}^{m}=50$ nT, $B_{z}^{m}=50$ nT). Blue and red curves represent the numerical result and the analytic result, respectively.

**Figure 3 sensors-23-03148-f003:**
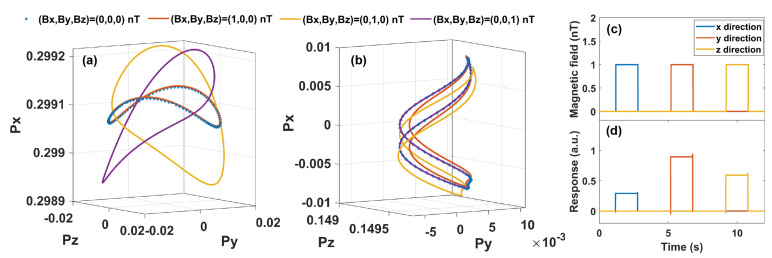
Response of the atomic polarization vector to the change in the static magnetic field under strong modulation. Trajectories of the polarization vector are presented in (**a**) for the atoms of P1 part and (**b**) for the atoms of P2 part within the period of ten complete cycles under different static magnetic fields. (**c**) Change in the static magnetic field. (**d**) Response of the whole atomic spin to the change in magnetic field in three directions.

**Figure 4 sensors-23-03148-f004:**
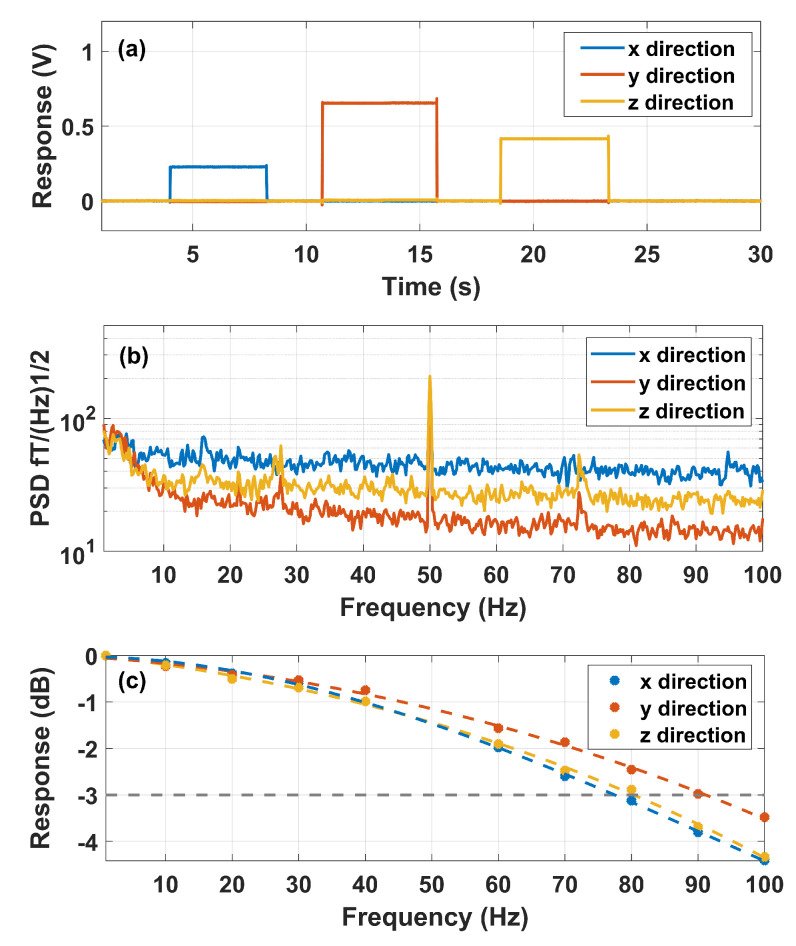
(**a**) Response of the AM to the changes in the magnetic field along three orthogonal directions. (**b**) Triaxial Sensitivity of the AM. (**c**) The frequency responses of the AM in x, y and z directions.

**Figure 5 sensors-23-03148-f005:**
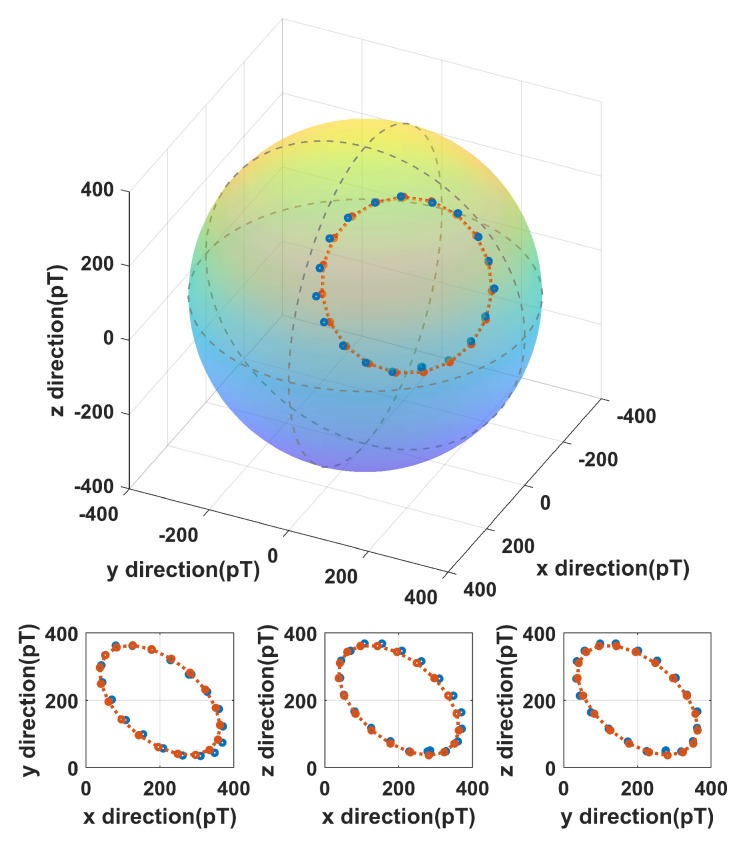
Comparison of the experimental and expected values when measuring magnetic fields of various orientations. Red dots present the expected values of applied magnetic fields, and blue dots present the values recorded by the vectorial AM. The projections of each dot on each coordinate plane are also presented.

**Figure 6 sensors-23-03148-f006:**
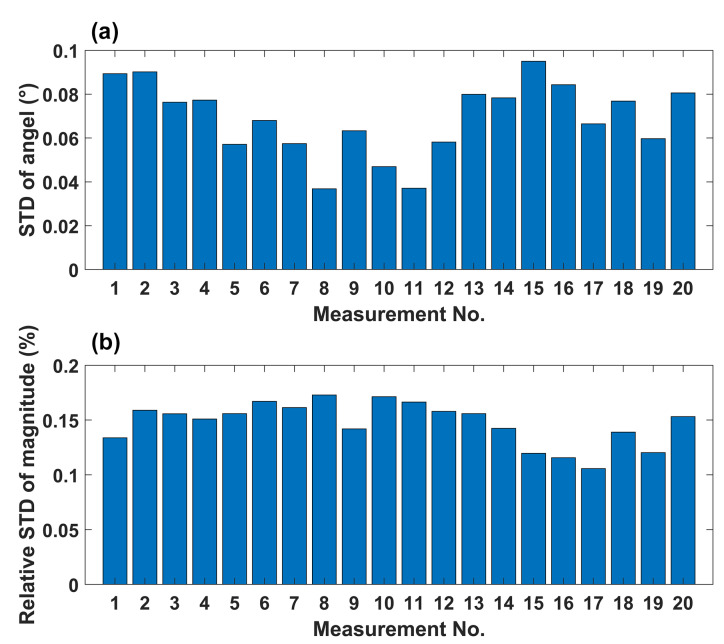
The standard deviation of 2000 measurements for each applied magnetic field. (**a**) Deviations of the orientation measurement. (**b**) Deviations of the magnitude measurement.

**Table 1 sensors-23-03148-t001:** Normalized responses in x, y and z direction by the crosstalk effect.

Normalized Response	X	Y	Z
Bx applied	1	0.0086	0.0224
By applied	0.0263	1	0.0059
Bz applied	0.0288	0.0022	1

## Data Availability

Not applicable.

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
