# Peer review of "An Integrated Single-Beam Three-Axis High-Sensitivity Magnetometer"

_sensors, 2023, doi:10.3390/s23063148_

Round 1

Reviewer 1 Report

The authors proposed a specifically designed alkali cell and modulation technique for tri-axis vector magnetic field measurement, requiring only one laser beam. And the cross-talks between the three axes are small. Such compact and simple-structure atomic magnetometers can be useful for some applications, such as biomedical applications. Here are some suggestions:

1. In the introduction, when the authors review vector magnetometry, they should briefly introduce how these approaches extract vector information, not just the limits of these approaches. There is a typo, “Seltzera” should be “Seltzer.” The section introduction might be section 1 instead of 0.

2. The parameters in Section: Experimental results need to be clarified. What are the amplitudes of the modulation magnetic field in x, y, and z? What are the modulation frequencies applied here?

3. In figure 4, the authors discussed the sensitivities of three-axis magnetic field measurements. It is necessary to include the frequency response of each axis. Is the power spectral density spectrum normalized based on the frequency responses?

Reviewer 2 Report

The paper "An integrated single-beam..." is complete and very interesting to read. It is a big step forward if all results can be translated into duplications of the device. Nice work!

The are a range of minor issues that need some consideration:

Introduction: "is proven to be little": Here and later some quantification of the crosstalk should be made.

Theory: "low-noise heater" - More detail please, how is it realised.

Theory: "optimization of the wavelength" - More explanation please, the wavelength is a stable value?

Principle of AM: "twice the modulation frequency" - Why are the authors sure that there is no 2f contribution from the y and z modulation? Because this could overlap with the 2f modulation in the x direction.

Principle of AM: "radio frequency field" - This is a matter of phrasing, if 2 kHz ai really radio frequency? Maybe just straight "modulation frequency"?

Principle of AM: "strong demodulation / weak demodulation" - These are the cases of strong/weak modulation. Demodulation is probably confusing.

Experimental results: Is strong or weak modulation used in the experiment? Why the choice?

Experimental results:  "triaxial-signal synchronous" - Please give some details about the electronics. Is it a digital output from the demodulation / lock-in calculation or is an analog output realized, which is then digitized?

Experimental results: Figure 5 is quite impressive, neverthess the projections on the coordinate planes are not easy to assess. Please  show the 3 projections as 2D plots.

Experimental results: Language improvement needed for "while with various".

Exp. results: "accuracy of the AM" - Please give a quantitative value.

Experimental results: Language improvement needed for "20 magnetic fields".
